# Multi-Granularity Cross-modal Alignment for Generalized Medical Visual Representation Learning

**Fuying Wang[1], Yuyin Zhou[2], Shujun Wang[3], Varut Vardhanabhuti[1], Lequan Yu[1]***

[1]The University of Hong Kong    [2]University of California, Santa Cruz    [3]University of Cambridge

`{fuyingw@connect., varv@, lqyu@}hku.hk`
`yzhou284@ucsc.edu`
`sw991@cam.ac.uk`

## Abstract

Learning medical visual representations directly from paired radiology reports has become an emerging topic in representation learning. However, existing medical image-text joint learning methods are limited by instance or local supervision analysis, ignoring disease-level semantic correspondences. In this paper, we present a novel **M**ulti-**G**ranularity **C**ross-modal **A**lignment (MGCA) framework for generalized medical visual representation learning by harnessing the naturally exhibited semantic correspondences between medical image and radiology reports at three different levels, *i.e.*, pathological region-level, instance-level, and disease-level. Specifically, we first incorporate the instance-wise alignment module by maximizing the agreement between image-report pairs. Further, for token-wise alignment, we introduce a bidirectional cross-attention strategy to explicitly learn the matching between fine-grained visual tokens and text tokens, followed by contrastive learning to align them. More important, to leverage the high-level inter-subject relationship semantic (*e.g.*, disease) correspondences, we design a novel cross-modal disease-level alignment paradigm to enforce the cross-modal cluster assignment consistency. Extensive experimental results on seven downstream medical image datasets covering image classification, object detection, and semantic segmentation tasks demonstrate the stable and superior performance of our framework.

## 1 Introduction

In recent decades, deep learning techniques have significantly advanced medical image understanding when large-scale labeled datasets are available [46, 20, 14, 9, 44]. However, assembling such large annotated data is expensive and time-consuming. As an alternative, learning directly from radiology reports accompanied by medical images becomes mainstream without any extra manual annotation, which aims to learn general medical vision representations from physicians' detailed medical records and then transfer the learned representations to downstream tasks. In the previous literature, image-text contrastive learning has achieved huge success for a wide range of medical downstream tasks [61, 27, 62] by predicting which radiology report goes with which medical image. Considering one limitation that pathologies usually occupy a small part of the whole image, Huang *et al.* [27] proposed attention-based contrastive learning strategy to learn local representations, which can capture fine-grained semantics in medical images to facilitate localized downstream tasks, *e.g.*, medical object detection and medical semantic segmentation.

As we can see in Figure 1, medical images and radiology reports naturally exhibit multi-granularity semantic correspondence at different levels, *e.g.*, disease-level, instance-level, and pathological region-level. However, existing image-text joint learning methods are limited to explore correspondence

---

*Corresponding author.

36th Conference on Neural Information Processing Systems (NeurIPS 2022).

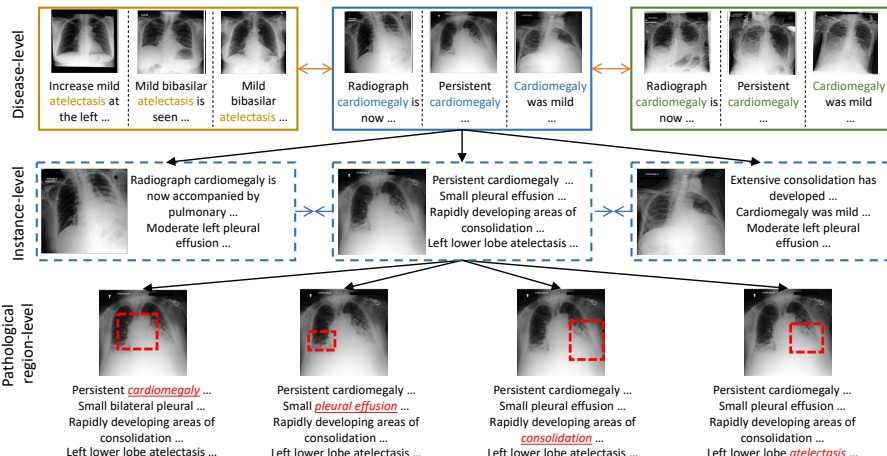

Figure 1: The multi-granularity (disease-level, instance-level, and pathological region-level) semantic correspondences across medical images and radiology reports.

supervision from only a part of levels [4, 24, 61, 27, 42], leading to inefficient usage of the valuable medical image/report information and insufficient representation capability of models. Thus, how to effectively leverage this intrinsic multi-granularity cross-modal correspondences between medical images and radiology reports from all three levels to enhance medical visual representation remains an open question.

In this paper, we present a novel **M**ulti-**G**ranularity **C**ross-modal **A**lignment (MGCA) framework to seamlessly leverage the multi-granular semantic correspondences between medical images and radiology reports for generalized medical visual representation learning. Specifically, we incorporate well-known instance-wise alignment, fine-grained token-wise alignment, and disease-level alignment via contrastive learning to enhance the generalizability of learned visual representations. Particularly, for token-wise alignment, we introduce an effective bidirectional cross-attention strategy to explicitly learn the soft matching between local image-text representations, and then adopt a cross-modal contrastive learning scheme to improve the sensitivity of local features. For disease-level alignment, we leverage the inter-subject relationship correspondence between medical images and radiology reports by enforcing the cross-modal cluster assignment consistency. By exploiting multi-granular cross-modal correspondences from three aspects, our MGCA framework has the ability to boost downstream tasks at both image and pixel levels, where only limited annotated data are required. We pre-train our MGCA framework with a large-scale medical image and report dataset, *i.e.*, MIMIC-CXR, and then validate our learned medical visual representations with seven downstream datasets, belonging to three medical tasks: image classification, object detection, and semantic segmentation. Experimental results demonstrate that our model achieves stable superior transfer performance even training with $1\%$ of training data, when compared with existing state-of-the-art medical image-text pre-training methods. Our code is in https://github.com/fuying-wang/MGCA.

## 2 Related Work

**Learning Medical Visual Representations from Reports** There are two mainstream paradigms to learn medical visual representations from report supervision. The first is to extract disease labels from radiology reports via human-designed rules [29, 31, 55] and then pre-train an image model for downstream tasks. However, defining such rules requires a lot of human labor and expert knowledge. Also, the trained models may be suboptimal, as the extracted labels are usually noisy [55]. The second focuses on leveraging vision-language contrastive learning to pre-train image and text encoders in an unsupervised manner [61, 27, 42, 62]. Supervised by naturally occurring of medical images and radiology reports, these methods demonstrate remarkable performance in various medical image downstream tasks (*e.g.*, image classification [61, 27], semantic segmentation [27], image-image retrieval [61], image-text retrieval [61, 27]). However, these methods only utilize partial correspondence supervision of cross-modal semantics, leading to suboptimal performance for downstream tasks.

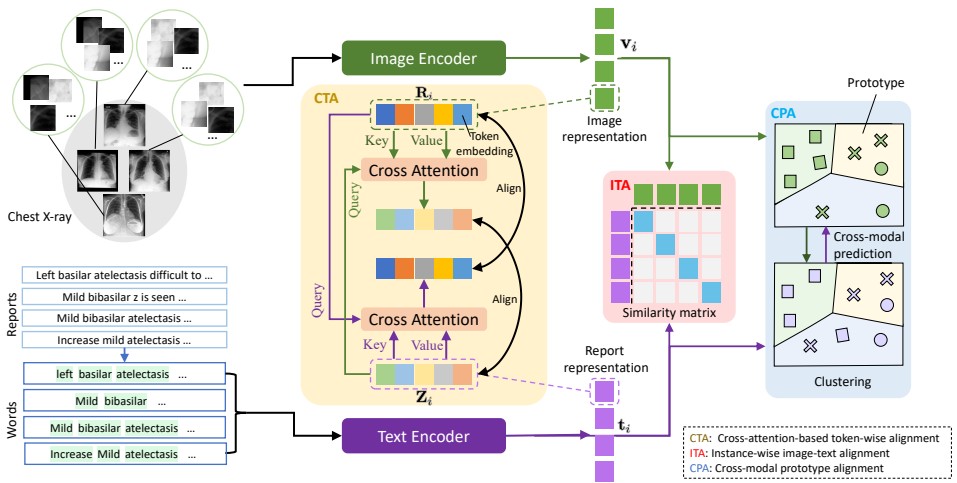

Figure 2: Illustration of our proposed multi-granularity cross-modal alignment framework. CTA, ITA, and CPA represent token-wise alignment, instance-wise alignment, and prototype (disease)-level alignment respectively. The green arrow represents information flow of visual features, while the purple arrow represents information flow of textural features.

**Contrastive Learning** Contrastive learning [22, 4, 24, 17] aims to learn an embedding space where positive instances stay close to each other, while negative pairs are far apart. One of the key technique is to find effective positive and negative pairs. To improve the efficiency of contrastive learning, some recent works proposed to predict one view's representations from another view [4, 19]. Moreover, [3, 23, 49, 17] brought the power of contrastive learning into medical image domain and have achieved substantial performance . We refer the readers to the survey [32, 30] for more details. Recently, some prototypical contrast learning approaches were proposed to exploit the prototype-level semantics in the dataset. For example, [35, 54, 21] proposed contrasting instance features with its paired prototype features, while [37, 2] proposed clustering-based methods to contrast prototype-prototype pairs. Different from these methods working on a single modality, we propose a cross-modal prototype alignment module to exploit the cross-modal prototype-level semantic consistency.

**Unsupervised Learning for Dense Prediction** Learning fine-grained semantic correspondence is essential for dense prediction tasks, *e.g.*, object detection [15, 38], semantic segmentation [6], *etc.*Recent studies have proposed various paradigms to tackle this problem [33, 28, 36, 11, 59]. However, most of these methods require a pre-trained object detector to generate proposals of interested objects. Dense contrastive learning is thus proposed to learn fine-grained visual representations without a pre-trained object detector [58, 57, 56, 3]. By optimizing a contrastive loss at pixel-level (or region-level) between two views of input images, these methods effectively learn the correspondence between local features, which significantly benefits dense prediction tasks.

## 3 Method

### 3.1 Overview

We aim to learn a generalized medical image representations from radiology reports to benefit various downstream medical image recognition tasks where annotated data is limited. Given a training set of $N$ image-report pairs $\mathcal{D} = \{(x_{v,1}, x_{t,1}), (x_{v,2}, x_{t,2}), ..., (x_{v,N}, x_{t,N})\}$, we utilize an image encoder $f_v$ (*e.g.*, ViT [12]) and a text encoder $f_t$ (*e.g.*, BERT [10]) to map $\mathcal{D}$ into the latent space $\mathcal{E} = \{(\mathbf{v}_1, \mathbf{t}_1), (\mathbf{v}_2, \mathbf{t}_2), ..., (\mathbf{v}_N, \mathbf{t}_N)\}$, where $\mathbf{v}_i = f_v(x_{v,i}), \mathbf{t}_i = f_t(x_{t,i})$. In detail, for the $i$-th image-report pair $(x_{v,i}, x_{t,i})$, the image encoder generates a sequence of encoded visual tokens $\mathbf{R}_i = \{\mathbf{r}_i^1, \mathbf{r}_i^2, ..., \mathbf{r}_i^S\}$ and a global image representation $\mathbf{v}_i$. Similarly, the text encoder generates a sequence of encoded text tokens $\mathbf{Z}_i = \{\mathbf{z}_i^1, \mathbf{z}_i^2, ..., \mathbf{z}_i^L\}$ and a global report representation $\mathbf{t}_i$. Here, $S$ and $L$ denote the total number of visual tokens and text tokens, respectively.

As illustrated in Figure 2, we design a novel multi-granularity cross-modal alignment framework for representation learning by exploiting the naturally exhibited multi-granularity cross-modal correspondences, *i.e.*, disease (prototype)-level, instance-level, and pathology region (token)-level, between

images and reports. Specifically, we incorporate an instance-wise image-text alignment (ITA) module to retain the cross-modal smoothness by maximizing the agreement between true image-report pairs $(\mathbf{v}_i, \mathbf{t}_i)$ versus random pairs. Meanwhile, to harness the benefits of naturally existing fine-grained correspondence between visual and text tokens, we also introduce a bidirectional cross-attention-based token-wise alignment (CTA) module to learn the soft matching between the visual token sequence $\mathbf{R}_i$ and text token sequence $\mathbf{Z}_i$, as well as align them via the contrastive learning. More important, we design a novel cross-modal prototype alignment (CPA) module to benefit the high-level semantic (*e.g.*, disease) understanding by capturing the inter-subject relationship correspondence between two modalities.

### 3.2  Multi-granularity Cross-modal Alignment

**Instance-wise Cross-modal Alignment**    As the core workhorses in our framework, we incorporate an instance-wise Image-Text Alignment (ITA) module to encourage the framework to map correct image-report pairs nearby in the latent space, while mapping random pairs far apart. Specifically, following common practice [4], we first use two non-linear projection layers ($g_v$ and $g_t$) to transform $\mathbf{v}_i$ and $\mathbf{t}_i$ into normalized lower-dimensional embeddings $\tilde{\mathbf{v}}_i \in \mathbb{R}^d$ and $\tilde{\mathbf{t}}_i \in \mathbb{R}^d$, respectively. Then, the cosine similarity of $i$-th image-report pair is denoted as:

$$sim(x_{v,i}, x_{t,i}) = \tilde{\mathbf{v}}_i^T \tilde{\mathbf{t}}_i, \text{where } \tilde{\mathbf{v}}_i = g_v(\mathbf{v}_i), \tilde{\mathbf{t}}_i = g_t(\mathbf{t}_i). \tag{1}$$

For $i$-th image-report pair $(x_{v,i}, x_{t,i})$ in a minibatch, we regard two modality data as queries and keys alternatively to learn the correct image-report pairings. This results in two symmetric temperature-normalized InfoNCE [43] losses (image-to-text contrastive loss and text-to-image contrastive loss) to maximally preserve the mutual information between the true pairs in latent space:

$$\ell_i^{\text{v2t}} = -\log \frac{\exp(sim(x_{v,i}, x_{t,i})/\tau_1)}{\sum_{k=1}^B \exp(sim(x_{v,i}, x_{t,k})/\tau_1)}, \quad \ell_i^{\text{t2v}} = -\log \frac{\exp(sim(x_{t,i}, x_{v,i})/\tau_1)}{\sum_{k=1}^B \exp(sim(x_{t,i}, x_{v,k})/\tau_1)}, \tag{2}$$

where $B$ is the batch size and $\tau_1$ is the instance-level temperature hyperparameter. The overall objective of our instance-wise cross-modal alignment is the average of the two losses:

$$\mathcal{L}_{\text{ITA}} = \frac{1}{2N} \sum_{i=1}^N (\ell_i^{\text{v2t}} + \ell_i^{\text{t2v}}), \tag{3}$$

where $N$ is the total number of image-report pairs.

**Token-wise Cross-modal Alignment**    Fine-grained information is more significant in medical field: pathologies only occupy a small portion of the whole image and only a few disease tags in a report depict the crucial medical condition. Considering that these important subtle clues are likely to be ignored when optimizing the global instance-wise representations, we introduce an effective bidirectional Cross-attention-based Token-wise Alignment (CTA) module to explicitly match and align the cross-modal local presentations between medical images and radiology reports.

Specifically, for the $i$-th image-report pair $(x_{v,i}, x_{t,i})$, the generated visual and text token embeddings will first be projected into normalized lower-dimensional embeddings, which results in $\tilde{\mathbf{R}}_i = \{\tilde{\mathbf{r}}_i^1, \tilde{\mathbf{r}}_i^2, ..., \tilde{\mathbf{r}}_i^S\}$ and $\tilde{\mathbf{Z}}_i = \{\tilde{\mathbf{z}}_i^1, \tilde{\mathbf{z}}_i^2, ..., \tilde{\mathbf{z}}_i^L\}$, where $\tilde{\mathbf{r}}_i \in \mathbb{R}^d, \tilde{\mathbf{z}}_i \in \mathbb{R}^d$. In order to conduct the token-wise alignment, we need to find the matching between visual and text tokens. Instead of directly computing the cosine similarity of different tokens [27], we propose to calculate the soft matching between generated visual and text tokens with the cross-attention mechanism [52, 5, 41]. Formally, for the $j$-th visual token embedding $\tilde{\mathbf{r}}_i^j$ in $i$-th image-report pair, we let $\tilde{\mathbf{r}}_i^j$ attend to all text token embeddings in $\tilde{\mathbf{Z}}_i$ and then calculate its corresponded cross-modal text embedding $\mathbf{o}_i^j$,

$$\mathbf{o}_i^j = \sum_{k=1}^N O(\alpha_i^{j2k}(V\tilde{\mathbf{z}}_i^k)), \; \alpha_i^{j2k} = \text{softmax}(\frac{(Q\tilde{\mathbf{r}}_i^j)^T(K\tilde{\mathbf{z}}_i^k)}{\sqrt{d}}), \tag{4}$$

where $Q \in \mathbb{R}^{d \times d}, K \in \mathbb{R}^{d \times d}, V \in \mathbb{R}^{d \times d}$ are learnable matrices. After that, we adopt a Local Image-to-text Alignment (LIA) loss $\mathcal{L}_{\text{LIA}}$ to pull $\tilde{\mathbf{r}}_i^j$ close to its cross-modal text embedding $\mathbf{o}_i^j$ but push $\tilde{\mathbf{r}}_i^j$ away from other cross-modal text embeddings, which maximizes the lower bound on

local cross-modal mutual information within each image-report pair [43]. Considering that different visual tokens have various importance (*e.g.*, visual tokens containing pathologies are obviously more important than those with irrelevant information), we further assign a weight $w_i^j$ to the $j$-th visual token when calculating the LIA loss. So the LIA loss can be formulated as:

$$\mathcal{L}_{\text{LIA}} = -\frac{1}{2NS} \sum_{i=1}^{N} \sum_{j=1}^{S} w_i^j (\log \frac{\exp(sim(\tilde{\mathbf{r}}_i^j, \mathbf{o}_i^j)/\tau_2)}{\sum_{k=1}^{S} \exp(sim(\tilde{\mathbf{r}}_i^j, \mathbf{o}_i^k)/\tau_2)} + \log \frac{\exp(sim(\mathbf{o}_i^j, \tilde{\mathbf{r}}_i^j)/\tau_2)}{\sum_{k=1}^{S} \exp(sim(\mathbf{o}_i^j, \tilde{\mathbf{r}}_i^k)/\tau_2)}), \tag{5}$$

where $\tau_2$ is the token-level temperature hyperparameter. Similar with the instance-level alignment, we also employ two symmetric InfoNCE losses by taking the visual token embedding $\tilde{\mathbf{r}}_i^j$ and its cross-modal text embedding $\mathbf{o}_i^j$ as queries, respectively. Note that we set $w_i^j$ as the last-layer attention weight from $j$-th visual token to [CLS] token averaged across multiple heads. Similarly, for $j$-th text token embedding $\tilde{\mathbf{z}}_i^j$ in $i$-th image-report pair, we also calculate a cross-modal image embedding $\hat{\mathbf{o}}_i^j$ with the same manner and construct a Local Text-to-image alignment (*LTA*) loss $\mathcal{L}_{\text{LTA}}$ by contrasting $\tilde{\mathbf{z}}_j^i$ with $\hat{\mathbf{o}}_i^j$. The final objective of our CTA module is the combination of *LIA* and *LTA* losses:

$$\mathcal{L}_{\text{CTA}} = \frac{1}{2}(\mathcal{L}_{\text{LIA}} + \mathcal{L}_{\text{LTA}}). \tag{6}$$

**Discussion** Note that our token-wise alignment is different from the local contrastive loss in [27]. Our module explicitly contrasts similarities between local tokens to maximize their mutual information and calculates per-token InfoNCE, whereas the local contrastive loss in [27] contrasts aggregated instance-level similarities and still calculates per-instance InfoNCE.

**Disease-level Cross-modal Alignment** Both ITA and CTA treat two samples as a negative pair as long as they are from different instances. Many pairs sharing the similar high-level semantics (*e.g.*, disease) are undesirably pushed apart in the embedding space. Therefore, we design a novel Cross-modal Prototype Alignment (CPA) module to harness the cross-modal inter-subject correspondences between medical images and reports.

For each image-report embedding pair $(\tilde{\mathbf{v}}_i, \tilde{\mathbf{t}}_i)$ in (1), we employ the iterative Sinkhorn-Knopp clustering algorithm [8] to acquire two soft cluster assignment codes $\mathbf{q}_{v,i} \in \mathbb{R}^K$ and $\mathbf{q}_{t,i} \in \mathbb{R}^K$, by individually assigning $\tilde{\mathbf{v}}_i$ and $\tilde{\mathbf{t}}_i$ into $K$ clusters. Meanwhile, we also pre-define $K$ trainable cross-modal prototypes as $\mathcal{C} = \{\mathbf{c}_1, ..., \mathbf{c}_K\}$, where $\mathbf{c}_k \in \mathbb{R}^d$. After that, we calculate the visual softmax probability $\mathbf{p}_{v,i} \in \mathbb{R}^K$ of the cosine similarities between $\tilde{\mathbf{v}}_i$ and all cross-modal prototypes in $\mathcal{C}$, and the text softmax probability $\mathbf{p}_{t,i} \in \mathbb{R}^K$ of the cosine similarities between $\tilde{\mathbf{t}}_i$ and all cross-modal prototypes in $\mathcal{C}$,

$$\mathbf{p}_{v,i}^{(k)} = \frac{\exp(\tilde{\mathbf{v}}_i^T \mathbf{c_k}/\tau_3)}{\sum_{k'} \exp(\tilde{\mathbf{v}}_i^T \mathbf{c'_k}/\tau_3)}, \quad \mathbf{p}_{t,i}^{(k)} = \frac{\exp(\tilde{\mathbf{t}}_i^T \mathbf{c_k}/\tau_3)}{\sum_{k'} \exp(\tilde{\mathbf{t}}_i^T \mathbf{c'_k}/\tau_3)}, \tag{7}$$

where $\tau_3$ is the prototype-level temperature parameter and $(k)$ indicates the $k$-th element of the vector. The cross-modal disease-level (*i.e.*, prototype) alignment is achieved by conducting *cross-modal prediction* and optimizing the following two cross-entropy losses:

$$\ell(\tilde{\mathbf{v}}_i, \mathbf{q}_{t,i}) = \sum_{k=1}^{K} \mathbf{q}_{t,i}^{(k)} \log \mathbf{p}_{v,i}^{(k)}, \quad \ell(\tilde{\mathbf{t}}_i, \mathbf{q}_{v,i}) = \sum_{k=1}^{K} \mathbf{q}_{v,i}^{(k)} \log \mathbf{p}_{t,i}^{(k)}. \tag{8}$$

Here, the *cross-modal prediction* is implemented by taking the soft text cluster assignment code $\mathbf{q}_{t,i}$ as "pseudo-label" to train the image representation $\tilde{\mathbf{v}}_i$ and taking the soft image cluster assignment code $\mathbf{q}_{v,i}$ as "pseudo-label" to train the report representation $\tilde{\mathbf{t}}_i$. Finally, the overall CPA loss is the average of two prediction losses over all the image-report pairs:

$$\mathcal{L}_{\text{CPA}} = \frac{1}{2N} \sum_{i=1}^{N} (\ell(\tilde{\mathbf{v}}_i, \mathbf{q}_{t,i}) + \ell(\tilde{\mathbf{t}}_i, \mathbf{q}_{v,i})). \tag{9}$$

### 3.3 Overall Objective

We train our MGCA framework with jointly optimizing the three cross-modal alignment modules, encouraging the network to learn discriminative and generalizable medical image representation. The overall training objective can be represented as:

$$\mathcal{L} = \lambda_1 * \mathcal{L}_{\text{ITA}} + \lambda_2 * \mathcal{L}_{\text{CTA}} + \lambda_3 * \mathcal{L}_{\text{CPA}}, \tag{10}$$

where $\lambda_1$, $\lambda_2$, and $\lambda_3$ are hyperparameters to balance three-level cross-modal alignments.

## 4 Experiments

We pre-train our MGCA framework on a large-scale medical image-report dataset and then evaluate the effectiveness of learned medical image representations on seven datasets from three important downstream tasks in medical imaging. In the following subsections, we first introduce the experimental setup of pre-training in Section 4.1 and three downstream tasks in Section 4.2. Then, we compare our proposed framework with state-of-the-art medical image-text pre-training methods and show the comparison results in Section 4.3-4.5. Finally, we analyze our framework in Section 4.6. More analysis results can be found in the Appendix.

### 4.1 Pre-Training Setup

**Dataset**  We pre-train our MGCA framework on the JPG version of **MIMIC-CXR 2.0.0** dataset [31]. We follow [61] to preprocess the dataset. We remove all lateral views from the dataset, as the downstream datasets only contain frontal-view chest images. Also, we extract the impression and finding sections from free-text reports to obtain detailed descriptions of medical diseases and remove reports which are empty or have less than 3 tokens, resulting in roughly $217k$ image-text pairs.

**Implementation Details**  Following [27], we use BioClinicalBERT [1] as the text encoder. We choose ViT-B/16 [12] as the image encoder backbone by default for unified modal architecture design. It is worth noting that our framework is model-agnostic to the image encoder backbone and we also report the results with ResNet50 [25] as image encoder backbone for fair comparison with other methods. We train our framework 50 epochs on 2 pieces of RTX 3090 GPUs with batch size of 144. The optimizer is AdamW [40] with learning rate of $2e-5$ and weight decay of $0.05$. We use a linear warmup with cosine annealing scheduler [39]. We initialize learning rate as $1e-8$ and warmup epoch as 20. Following the practice in contrastive learning [4, 24], the dimension $d = 128$ and the temperature hyperparameters are $\tau_1 = 0.1, \tau_2 = 0.07, \tau_3 = 0.2$. The number of prototypes is $K = 500$. We set $\lambda_1 = 1, \lambda_2 = 1, \lambda_3 = 1$. More pre-training details can be found in the Appendix.

### 4.2 Downstream Tasks and Experimental Setup

**Medical Image Classification**  We conduct medical image classification on three representative datasets: (1) **CheXpert** [29], which contains $191,229$ frontal chest radiographs. The task is to classify each image into 5 individual binary labels: *atelectasis*, *cardiomegaly*, *consolidation*, *edema*, and *pleural effusion*. Following [61, 27], we hold out the expert-labeled validation set as test data and randomly select $5,000$ radiographs from training data for validation. (2) **RSNA** Pneumonia [47]. We use the stage 2 version, which contains around $29,700$ frontal view chest radiographs. The task is a binary classification, *i.e.*, classifying each chest image into *normal* or *pneumothorax positive*. Following [27], we manually split the dataset into training, validation, and test set with $70\%/15\%/15\%$ ratio. (3) **COVIDx** [53], which contains over $30k$ CXR images from a multinational cohort of over $16,600$ patients. This dataset contains $16,490$ positive COVID-19 images from over $2,800$ patients. We use the latest version 6 of this dataset. The task is a three-class classification, *i.e.*, classifying each radiograph into *COVID-19*, *non-COVID pneumonia* or *normal*. We use the original validation dataset as test data and manually split $10\%$ of original training set for validation.

Following the previous work [27], we use the *Linear Classification* setting to evaluate the transferability of our pre-trained image encoder, *i.e.*, freezing the pre-trained ViT/ResNet-50 image encoder and only training a randomly initialized linear classification head for the downstream classification task. Also, we evaluate our model with $1\%$, $10\%$, and $100\%$ training data on each classification dataset to further verify the data efficiency of our method. We report area under the ROC curve (AUROC) on CheXpert and RSNA and acuracy (ACC) on COVIDx-v6 as the evaluation metric following [61].

Table 1: Linear classification results on CheXpert, RSNA and COVIDx with $1\%, 10\%, 100\%$ training data. Area under ROC curve (AUROC [%]) are reported for CheXpert and RSNA dataset, and accuracy (ACC [%]) is reported for COVIDx dataset. The best and second-best results are highlighted in red and blue, respectively.

| Method | CheXpert (AUC) | | | RSNA (AUC) | | | COVIDx (ACC) | | |
|---|---|---|---|---|---|---|---|---|---|
| | 1% | 10% | 100% | 1% | 10% | 100% | 1% | 10% | 100% |
| Random Init | 56.1 | 62.6 | 65.7 | 58.9 | 69.4 | 74.1 | 50.5 | 60.3 | 70.0 |
| ImageNet Init | 74.4 | 79.7 | 81.4 | 74.9 | 74.5 | 76.3 | 64.8 | 78.8 | 86.3 |
| *pre-trained on CheXpert* | | | | | | | | | |
| DSVE [13] | 50.1 | 51.0 | 51.5 | 49.7 | 52.1 | 57.8 | - | - | - |
| VSE++ [16] | 50.3 | 51.2 | 52.4 | 49.4 | 57.2 | 67.9 | - | - | - |
| GLoRIA [27] | 86.6 | 87.8 | 88.1 | 86.1 | 88.0 | 88.6 | 67.3 | 77.8 | 89.0 |
| *pre-trained on MIMIC-CXR* | | | | | | | | | |
| Caption-Transformer [7] | 77.2 | 82.6 | 83.9 | - | - | - | - | - | - |
| Caption-LSTM [60] | 85.2 | 85.3 | 86.2 | - | - | - | - | - | - |
| Contrastive-Binary [50][48] | 84.5 | 85.6 | 85.8 | - | - | - | - | - | - |
| ConVIRT [61] | 85.9 | 86.8 | 87.3 | 77.4 | 80.1 | 81.3 | 72.5 | 82.5 | 92.0 |
| GLoRIA-MIMIC [27] | 87.1 | 88.7 | 88.0 | 87.0 | 89.4 | 90.2 | 66.5 | 80.5 | 88.8 |
| **MGCA(Ours, ResNet-50)** | 87.6 | 88.0 | 88.2 | 88.6 | 89.1 | 89.9 | 72.0 | 83.5 | 90.5 |
| **MGCA(Ours, ViT-B/16)** | 88.8 | 89.1 | 89.7 | 89.1 | 89.9 | 90.8 | 74.8 | 84.8 | 92.3 |

Table 2: Object detection results (mAP [%]) on RSNA and Object CXR. Each dataset is fine-tuned with $1\%, 10\%, 100\%$ training data. Best results are in boldface. "-" means mAP is smaller than $1\%$.

| Method | RSNA | | | Object CXR | | |
|---|---|---|---|---|---|---|
| | 1% | 10% | 100% | 1% | 10% | 100% |
| Random | 1.00 | 4.00 | 8.90 | - | 0.49 | 4.40 |
| ImageNet | 3.60 | 8.00 | 15.7 | - | 2.90 | 8.30 |
| ConVIRT [61] | 8.20 | 15.6 | 17.9 | - | 8.60 | 15.9 |
| GLoRIA [27] | 9.80 | 14.8 | 18.8 | - | 10.6 | 15.6 |
| GLoRIA-MIMIC [27] | 11.6 | 16.1 | 24.8 | - | 8.90 | 16.6 |
| **MGCA (Ours)** | **12.9** | **16.8** | **24.9** | - | **12.1** | **19.2** |

**Medical Object Detection**    We evaluate the localized performance of pre-trained image encoder on two object detection tasks: (1) **RNSA** Pneumonia [47] contains 29700 frontal view radiograph. The task is to predict bounding boxes indicating evidence of pneumonia. We randomly split the original training set into $16,010/5,337/5,337$ for training/validation/testing. (2) **Object CXR [26]** contains $9,000$ frontal-view chest X-rays with detection targets for foreign objects. We use the original development set as test set $(1,000)$ and randomly split the original training set into training $(6,400)$ and validation $(1,600)$ sets.

We evaluate the detection performance by YOLOv3 [45] frozen setting, *i.e.*, using the pre-trained ResNet-50 image encoder as a frozen backbone of a YOLOv3 model and only fine-tuning the non-backbone layers. Similarly, we fine-tune the model by $1\%, 10\%$ and $100\%$ training data to evaluate the data efficiency. Mean Average Precisions (mAP) are reported as evaluation metric with IOU thresholds $0.4, 0.45, 0.5, 0.55, 0.6, 0.65, 0.7, 0.75$.

**Medical Semantic Segmentation**    We also evaluate the performance of our framework for medical semantic segmentation on SIIM and RNSA datasets: (1) **SIIM** Pneumothorax [18] dataset contains 12047 chest radiographs with manually annotated segmentation mask of pneumothorax. Following [27], train/validation/test split respectively constitutes $70\%/30\%/30\%$ of original dataset. (2) **RNSA** Pneumonia [47] is with the same split protocol as object detection task. We convert object detection ground truths into masks for semantic segmentation.

Following [27], we evaluate the segmentation performance by U-Net [46] fine-tuning protocol. We use the pre-trained ResNet-50 image encoder as a frozen encoder backbone of U-Net and train the decoder portion using $1\%, 10\%$ and $100\%$ training data. Dice scores are reported to evaluate the segmentation performance. The other downstream experimental setup can be found in the Appendix.

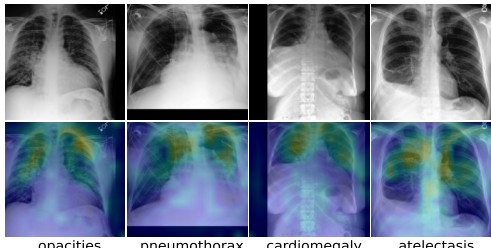
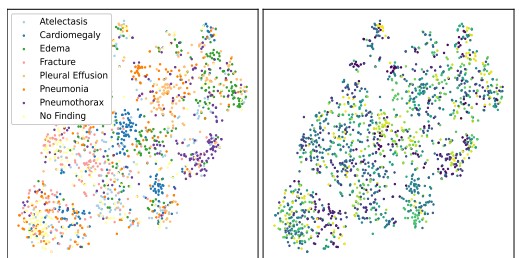

| opacities | pneumothorax | cardiomegaly | atelectasis |

Figure 3: Visualization of learned token correspondence by our MGCA. Highlighted pixels represent higher activation weights by corresponding word.

Figure 4: t-SNE visualizations of encoded image representations. Colors indicate the ground truth disease types and cluster assignment in left and right sub-figures.

Table 3: Semantic segmentation results (Dice [%]) on SIIM and RSNA. Each dataset is fine-tuned with $1\%, 10\%, 100\%$ training data. Best results of each setting are in boldface.

| Method | SIIM | | | RSNA | | |
| --- | --- | --- | --- | --- | --- | --- |
| | 1% | 10% | 100% | 1% | 10% | 100% |
| Random | 9.00 | 28.6 | 54.3 | 6.90 | 10.6 | 18.5 |
| ImageNet | 10.2 | 35.5 | 63.5 | 34.8 | 39.9 | 64.0 |
| ConVIRT[61] | 25.0 | 43.2 | 59.9 | 55.0 | 67.4 | 67.5 |
| GLoRIA[27] | 35.8 | 46.9 | 63.4 | 59.3 | 67.5 | 67.8 |
| GLoRIA-MIMIC [27] | 37.4 | 57.1 | 64.0 | 60.3 | **68.7** | 68.3 |
| **MGCA (Ours)** | **49.7** | **59.3** | **64.2** | **63.0** | 68.3 | **69.8** |

## 4.3 Results

**Results on Medical Image Classification** Table 1 reports the results on three classification tasks. The results of other methods on CheXpert and RSNA are from original papers[2]. According to the pre-training dataset, we group the existing pre-training methods into two categories : *pre-trained on CheXpert* and *pre-trained on MIMIC-CXR*. As GLoRIA only reports results with pre-training on CheXpert dataset, we also reimplement their method with pre-training on MIMIC-CXR dataset (GLoRIA-MIMIC) for a fair comparison. It is observed that our MGCA with ViT-B/16 backbone shows the best performance in all nine settings, outperforming state-of-the-art GLoRIA [27] and ConVIRT [61]. With the same ResNet-50 image encoder backbone, our framework also achieves second-best performance on four settings and competitive performance on the remaining five settings, showing the effectiveness of our framework. When fine-tuning with $1\%$ proportion of data, our MGCA with ViT-B/16 backbone outperforms GLoRIA-MIMIC with $1.7\%$ AUROC on CheXpert, $2.1\%$ AUROC on RSNA dataset and $8.3\%$ ACC on COVIDx dataset, showing larger improvement than other methods and also indicating the data efficiency of our method.

**Results on Medical Object Detection** Table 2 reports the object detection performance on RSNA and Object CXR datasets. All methods adopt the same ResNet-50-YOLOv3 architecture. It is observed that under each setting, our MGCA outperforms ConVIRT, GLoRIA, and GLoRIA-MIMIC by a large margin. Importantly, our model shows superior detection performance when fine-tuning on $1\%$ training data, indicating that multi-granularity semantic alignment benefits the image encoder to learn more discriminative localized representations.

**Results on Medical Semantic Segmentation** Table 3 shows the semantic segmentation performance on SIIM and RSNA datasets with the same ResNet50-U-Net architecture. Compared with GLoRIA, GLoRIA-MIMIC, and ConVIRT, our MGCA shows higher dice scores on five over six settings. When training with $1\%$ portion of data, our MGCA achieves $12.3\%$ and $2.7\%$ Dice improvement than GLoRIA-MIMIC on SIIM and RSNA segmentation tasks, respectively. This comparison further validates the data efficiency of our method when transferring into dense prediction tasks.

---

[2]The results of ConVIRT [61] on RSNA dataset are from the reimplemented results in [27] as the RSNA dataset are updated by the organizer.

Table 4: Ablation study of our framework on linear classification (CheXpert and RSNA) and semantic segmentation (SIIM) settings. We report Area under ROC curve (AUROC [%]) on CheXpert and RSNA datasets, and (Dice [%]) on SIIM dataset. Best results of each setting are in boldface.

| Training tasks | | | CheXpert (AUC) | | | RSNA (AUC) | | | SIIM (Dice) | | |
|---|---|---|---|---|---|---|---|---|---|---|---|
| ITA | CTA | CPA | 1% | 10% | 100% | 1% | 10% | 100% | 1% | 10% | 100% |
| ✓ | | | 87.6 | 88.2 | 88.5 | 88.4 | 89.5 | 90.5 | 25.0 | 43.2 | 59.9 |
| ✓ | ✓ | | 88.3 | 88.9 | 89.1 | 88.9 | 89.8 | 90.7 | 47.6 | 54.4 | 61.3 |
| ✓ | | ✓ | 88.5 | 88.9 | 89.0 | 88.6 | 89.2 | 90.4 | 37.4 | 46.7 | 55.0 |
| ✓ | ✓ | ✓ | **88.8** | **89.1** | **89.7** | **89.1** | **89.9** | **90.8** | **49.7** | **59.3** | **64.2** |

Table 5: Results of natural VLP pre-trained models on linear classification setting.

| | CheXpert (AUC) | | | RSNA (AUC) | | |
|---|---|---|---|---|---|---|
| | 1% | 10% | 100% | 1% | 10% | 100% |
| BLIP [34] | 69.1 | 74.9 | 77.7 | 53.7 | 82.0 | 84.1 |
| **MGCA (Ours)** | **88.8** | **89.1** | **89.7** | **89.1** | **89.9** | **90.8** |

## 4.4 Analysis of Our Framework

**Visualization** To better understand the behaviour of our MGCA framework, we visualize the learned local correspondence of radiographs and medical reports in Figure 3. Our MGCA learns meaningful local correspondence between visual tokens and text tokens, which is helpful for the local discriminative feature learning. Moreover, we select 1600 medical images, each with one excluded abnormality, and present t-SNE plots [51] in Figure 4 to visualize image embeddings. The colors represent the ground truth and cluster assignment in left and right sub-figures. It is observed that our multi-modal prototypes can learn reasonable disease-level semantic information.

**Ablation Study of Component Design** Table 4 shows the ablation study results on two settings: medical image classification on CheXpert and RSNA datasets with ViT-B/16 as the image encoder backbone, and medical semantic segmentation on SIIM dataset with ResNet50 as the image encoder backbone. It is observed that CTA and CPA modules can both improve the classification performance, indicating that token-level alignment and prototype-level alignment facilitate the image encoder to learn more generalizable representations for downstream tasks. When combining CTA and CPA, we can obtain further improvement on all datasets, indicating that the benefits of CTA and CPA are complementary. According to the results on SIIM dataset, we notice that CTA and CPA can both improve semantic segmentation performance when combined with ITA. Interestingly, CTA improves a larger margin on the semantic segmentation performance than CPA, which further elaborates that CTA is helpful to learn fine-grained information. When we train ITA, CTA, and CPA jointly, it achieves the best performance.

**Results of Natural Vision-Language Pre-trained Model** Table 5 shows the results of fine-tuning the state-of-the-art natural Vision-Language Pre-trained (VLP) model BLIP [34], which is pre-trained on 14M image-text pairs. Due to the large domain discrepancy between natural image-text and medical image-text, directly transferring the pre-trained BLIP model to the downstream medical image tasks leads to inferior performance. This comparison indicates that pre-training on medical image-text datasets is necessary for capturing useful medical prior knowledge.

**Analysis of Error Bars** Table 6 shows error bars of our method on linear classification setting with ViT-B/16 as the image encoder backbone. We re-run each downstream task three times and calculate the mean and standard deviations. It is observed that the error bars are relatively small while comparing against other methods, which shows that our proposed method performs stably in these downstream tasks.

Table 6: Error bar of our methods on linear classification setting.

| | 1% | 10% | 100% |
|---|---|---|---|
| CheXpert (AUC) | $88.7 \pm 0.18$ | $89.13 \pm 0.16$ | $89.5 \pm 0.21$ |
| RSNA (AUC) | $89.03 \pm 0.11$ | $89.92 \pm 0.14$ | $90.77 \pm 0.04$ |
| COVIDX (ACC) | $73.9 \pm 0.64$ | $84.75 \pm 0.22$ | $92.85 \pm 0.50$ |

# 5   Discussion and Conclusion

This work presents MGCA, a multi-granularity cross-modal alignment framework for learning generalized medical visual representations from free-text radiology reports. By harnessing the naturally exhibited multi-granularity semantic correspondences across medical images and reports, our framework can learn generalized and discriminative medical image presentations for versatile downstream tasks to reduce the annotation burden. Extensive experimental results on seven downstream datasets demonstrate that our framework achieves substantial performance with limited annotated data.

**Limitations and Future Work** As our work mainly focuses on medical visual representation learning, we did not conduct experiments on image-image or image-text retrieval downstream tasks, which can be regarded as a limitation of our work. Our current framework learns the multi-granularity cross-modal alignment in parallel. In future work, we would like to explore how to leverage the multi-granularity correspondence in a holistic manner. Moreover, this paper mainly investigates the discrimination-based image-text pre-training. We also plan to extend our framework as the integration of discrimination-based and generation-based pre-training methods for medical image and text learning.

**Social Impacts** Our MGCA provides a promising solution to automatically diagnose abnormality of chest X-rays with limited annotated data, which can assist in reducing the workload of radiologists and promote the health in poor area. On the other hand, medical data (*e.g.*, chest X-rays, radiology reports *etc.*) may contain unintended private information or harmful texts, and we highly recommend users conduct a careful analysis of data before employing our model into practical applications.

# Acknowledgement

We gratefully thank HK HADCL for the support of this project. The work described in this paper is supported by HKU Seed Fund for Basic Research (Project No. 202009185079 and 202111159073).

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
