# Multi-Granularity Cross-modal Alignment for Generalized Medical Visual Representation Learning (Supplementary Material)

**Fuying Wang**[1]**, Yuyin Zhou**[2]**, Shujun Wang**[3]**, Varut Vardhanabhuti**[1]**, Lequan Yu**[1]*

[1]The University of Hong Kong    [2]University of California, Santa Cruz    [3]University of Cambridge

{fuyingw@connect., varv@, lqyu@}hku.hk

yzhou284@ucsc.edu

sw991@cam.ac.uk

## Roadmap of Appendix

The Appendix is organized as follows. We first provide more implementation details of pre-training and downstream tasks in Section A and Section B, respectively. Additional experimental results are in Section C. Last, we present additional visualization results in Section D.

## A  More Implementation Details of Pre-Training

### A.1  Data Preprocessing

We use JPG[2] version of MIMIC-CXR [12] dataset to pre-train our MGCA. For each image, we resize the larger size to 256 and pad zeros on the smaller side, which results in the image size of $256 \times 256$. During training, we randomly crop an $224 \times 224$ image and normalize it into range $[0, 1]$, and then feed it into the image encoder.

The medical reports are in the original MIMIC-CXR[3] dataset. Table 1 shows an example of radiology report. The impression/findings section of the reports can be considered as the "conclusion" of the radiologist. We use the open-source mimic-cxr repository[4] to extract impression and findings for each report. Following [9], we pick out sequences of alphanumeric characters and drop all other characters and symbols for all reports, and remove reports which contain less than 3 tokens. We tokenize each report using WordPiece tokenizer implemented by BioClinicalBERT [1] [5].

### A.2  Image and Text Encoders

**Image Encoder**    We use ViT-B/16 [5] as the image encoder by default, which is initialized with weights pre-trained on ImageNet-1k [16]. Following common practice in ViT [5], we split the radiograph with patch size $16 \times 16$, which results in 196 visual tokens for each image. Then, we prepend a learnable embedding ([CLS] token embedding) to the sequence of embedded patches, and feed them into the vision transformer. The embedding of [CLS] token in the last layer represents image-level feature (768-d vector), and the embeddings of patches in the last layer represent visual token-level features $f \in \mathbb{R}^{768 \times 196}$. The embedding dimension of ViT-B/16 is 768.

---

*Corresponding author.

[2]https://physionet.org/content/mimic-cxr-jpg/2.0.0/

[3]https://physionet.org/content/mimic-cxr/2.0.0/

[4]https://github.com/MIT-LCP/mimic-cxr

[5]https://huggingface.co/emilyalsentzer/Bio_ClinicalBERT

36th Conference on Neural Information Processing Systems (NeurIPS 2022).

Table 1: Example of a radiology report in MIMIC-CXR [12] dataset.

| |
|---|
| **EXAMINATION:** CHEST (PA AND LAT) |
| **INDICATION:** ___F with new onset ascites  eval for infection |
| **TECHNIQUE:** Chest PA and lateral |
| **COMPARISON:** None |
| **FINDINGS:**
There is no focal consolidation, pleural effusion or pneumothorax.
Bilateral nodular opacities that most likely represent nipple shadows.
The cardiomediastinal silhouette is normal.
Clips project over the left lung, potentially within the breast.
The imaged upper abdomen is unremarkable.
Chronic deformity of the posterior left sixth and seventh ribs are noted.
**IMPRESSION:**
No acute cardiopulmonary process. |

For fair comparison, we also conduct experiments on image encoder with ResNet-50 [7] backbone. The ResNet-50 is implemented by torchvision library[6] with weights initialized by ImageNet [16]. We follow [9] to upsample the input image from $224 \times 224$ into $299 \times 299$ and feed it into the ResNet-50. The output of the final adaptive average pooling layer is image-level feature (2048-d vector), and feature maps $f \in \mathbb{R}^{1024 \times 19 \times 19}$ of 3-rd bottleneck building block are token-level features. Then, we reshape the token-level features into $f \in \mathbb{R}^{1024 \times 361}$, where 361 is the total number of image regions.

**Text Encoder** The text encoder is a 6-layer $\text{BERT}_{\text{base}}$ [4] model, which is initialized by the first 6 layers of a BioClinicalBERT [1]. Note that we also tried using an original (12-layer) $\text{BERT}_{\text{base}}$ model as the text encoder, but did not find substantial improvement. We use impression and findings sections of each report to obtain detailed descriptions. We truncate each report or pad it with [PAD] tokens to make sure there are 112 text tokens in the report. The embedding of [CLS] token in the last layer represents the report-level feature (768-d vector), and embeddings of word tokens in the last layer represent text token-level features $f \in \mathbb{R}^{768 \times 112}$. Note that we also experimented with aggregating the last 4 layers' embeddings to obtain report-level feature and text token-level features, but did not find substantial improvement.

### A.3 Other Implementation Details

As shown in SimCLR [2], projection layer significantly improves contrastive learning performance. Thus, we also design two instance-level project layers (for images and reports) and two token-level nonlinear projection layers (for patches and words) to obtain better cross-modal alignment. The instance-level projection layer is a two-layer MultiLayer Perceptron (MLP) with Batch Normalization [10] and ReLU activation function. Additionally, we use a frozen Batch Normalization layer after the MLP to obtain instance-level embeddings. The token-level projection layer has the same architecture as the instance-level project layer, except that we replace all linear layer with 1-D convolution with kernel size 1, stride 1 and padding 0.

We pre-train MGCA for 50 epochs and early stop if total validation loss does not decrease after 5 straight runs. Then, we save the checkpoint that achieves the lowest validation loss for downstream tasks. The Cosine Annealing with Warmup scheduler is implemented by pytorch-cosine-annealing-with-warmup repository [7]. For efficiency, we employ mixed-precision (16-bit) training implemented by Pytorch Lightning[8]. The total training time on the MIMIC-CXR dataset is about 1 day on two RTX 3090 GPUs for readers' reference.

---

[6]`https://pytorch.org/vision/stable/models.html`
[7]`https://github.com/katsura-jp/pytorch-cosine-annealing-with-warmup`
[8]`https://pytorch-lightning.readthedocs.io/en/latest/guides/speed.html#mixed-precision-16-bit-training`

# B   More Implementation Details of Downstream Tasks

## B.1   Training Details of Classification Task

Except for fine-tuning $100\%$ portion of CheXpert [11] where we use batch size of 96, we use batch size 48 for the rest linear classification settings. Similar to preprocessing images of MIMIC-CXR, we resize the larger size to 256 and pad zeros on the smaller side, which results in the image size of $256 \times 256$. Then, we randomly crop (for training) or centered crop (for validation and test) an $224 \times 224$ image and normalize it into range $[0, 1]$ as the input of classifier.

In linear classification setting, we freeze the pre-trained image encoder (ResNet-50 or ViT-B/16), and only train the classification head (a randomly initialized linear layer). The optimizer is AdamW [13] with learning rate $5e - 4$ and weight decay $1e - 6$. We fine-tune the image classifier for 50 epochs and early stop when validation loss does not decrease for 10 straight runs. Then we save the checkpoint model with the lowest validation loss for test.

## B.2   Training Details of Detection Task

Except for fine-tuning $1\%$ portion of RSNA and Object-CXR [8] datasets with batch size of 8, we use batch size 16 for other object detection tasks. No data augmentation is used. We resize each image into $224 \times 224$ and normalize it into range $[0, 1]$, and then feed it into the object detection model.

The optimizer is AdamW with learning rate $5e - 4$ and weight decay $1e - 6$. We do not use learning rate scheduler. Note that we use the PyTorch implementation in this repository[9] and replace Darknet-53 backbone with a pre-trained ResNet-50. Before training, we freeze the image encoder and randomly initialize other layers. Three-stage features are extract from the 2-nd, 3-rd, 4-th bottleneck building block for predicting bounding boxes. The anchors are the same as the original paper [14] but we rescale them according to input image size $224 \times 224$. We fine-tune the object detection model for 50 epochs and early stop when validation loss does not decrease for 10 straight runs, and save the checkpoint model with the lowest validation loss for test.

## B.3   Training Details of Segmentation Task

We evaluate the segmentation performance of our MGCA on two datasets: SIIM Pneumothorax [6] Dataset and RSNA Pneumonia dataset. The data preprocessing procedure follows [9]. For RSNA dataset, we generate the mask of pneumonia regions according to the bounding boxes. Specifically, we resize the images and masks into size $512 \times 512$. For augmentation of training set, we apply ShiftScaleRotate provided by the albumentations[10] python library, which includes random affine transforms of translation, scaling, and rotation. Detailed parameters are: rotation limit is 10, scale limit is 0.1, and augmentation probability is 0.5. Then, we normalize the image into range $[0, 1]$, and feed it into the semantic segmentation model.

To evaluate the semantic segmentation performance of pre-trained **ResNet-50**, we use the U-Net [15] architecture with a ResNet-50 encoder implemented by Sementation-Models-PyTorch library[11]. During training, we use the AdamW optimizer with learning rate $5e - 4$ and weight decay $1e - 6$. Following [9], we use a combined loss of $\alpha \times \text{FocalLoss} + \text{DiceLoss}$ and set $\alpha = 10$. We fine-tune the semantic segmentation model for 50 epochs and early stop when validation loss does not decrease for 10 straight runs, and save the checkpoint model with the lowest validation loss for test.

To evaluate semantic segmentation performance of pre-trained **ViT-B/16**, we use the SETR-PUP (progressive upsample) architecture in [18] by replacing the encoder with pre-trained ViT. Our implementation is based on this repository[12]. Different from ResNet-50 variant, we resize each image into size $224 \times 224$ and feed it into SETR-PUP model. In this setting, we freeze the pre-trained image encoder and only train decoder portion. Loss function and other training hyperparameters as the same as the ResNet-50 U-Net fine-tuning setting.

---

[9]https://github.com/BobLiu20/YOLOv3_PyTorch
[10]https://albumentations.ai/
[11]https://github.com/qubvel/segmentation_models.pytorch
[12]https://github.com/920232796/SETR-pytorch

# C  Additional Analysis Results

## C.1  Segmentation Results of MGCA with ViT backbone

Table 2 compares the semantic segmentation performance of our MGCA with ResNet-50 and ViT-B/16 backbone. We report the dice score (%) training with 1%, 10%, and 100% portion on SIIM and RSNA datasets. The results show that ViT-B/16 backbone outperforms ResNet-50 counterpart in 5 out of 6 settings, which is the similar observations for classification task. This finding is also consistent with the common belief that Transformer might be an effective architecture to bridge vision and language domain.

Table 2: Semantic segmentation results (Dice [%]) of MGCA with different image encoder backbones.

| Backbone | SIIM | | | RSNA | | |
| --- | --- | --- | --- | --- | --- | --- |
| | 1% | 10% | 100% | 1% | 10% | 100% |
| ResNet-50 | **49.7** | 59.3 | 64.2 | 63.0 | 68.3 | 69.8 |
| ViT-B/16 | 49.0 | **64.7** | **66.4** | **66.2** | **71.3** | **73.6** |

## C.2  Results of Self-supervised Visual Learning on MIMIC

We also compare our MGCA with a popular image-only self-supervised learning method: MoCo v2 [3]. The learning objective of Moco v2 is to make positive pairs (same image with different augmentation) similar, while make negative pairs (different images) dissimilar. The training data is the same set of radiographs in MIMIC-CXR dataset. We use the implementation of Moco v2 in Lightning_bolts[13] repository. The data augmentation techniques are the same as the original paper [3]. Other training details (*e.g.*, optimizer, learning rate, scheduler, batch size, epochs, early stop. *etc.*) are the same as MGCA. The results show that our MGCA outperforms Moco v2 by a large margin, and it is worth studying medical image and report cross-modal pre-training methods.

Table 3: Comparison of our method and image-only contrastive learning methods.

| Method | CheXpert (AUC) | | | RSNA (AUC) | | |
| --- | --- | --- | --- | --- | --- | --- |
| | 1% | 10% | 100% | 1% | 10% | 100% |
| Moco v2 [3] | 81.6 | 84.7 | 84.9 | 85.0 | 87.4 | 88.2 |
| MGCA (ours) | **88.8** | **89.1** | **89.7** | **89.1** | **89.9** | **90.8** |

## C.3  More Ablation Analysis

We provide more ablation results in this subsection. As illustrated in the main text, we use image encoder with ViT-B/16 backbone to conduct ablation study. Detailed analysis are as follows.

**Weights of Three Learning Objectives**  Table 4 shows the effect of weights of three-level alignment objectives ($\lambda_1$, $\lambda_2$ and $\lambda_3$) on linear classification performance. We fix the weight of ITA to 1 and tune the weights of the other two losses (CTA and CPA). According to the table, $\lambda_2 = 1, \lambda_3 = 1$ generally achieves the best performance in 5 out of 6 settings and our framework is not very sensitive to different $\lambda$ values.

Table 4: Ablation results of hyperparameters $\lambda_1$, $\lambda_2$ and $\lambda_3$.

| Loss weights | | | CheXpert (AUC) | | | RSNA (AUC) | | |
| --- | --- | --- | --- | --- | --- | --- | --- | --- |
| $\lambda_1$ | $\lambda_2$ | $\lambda_3$ | 1% | 10% | 100% | 1% | 10% | 100% |
| 1 | 0.5 | 0.5 | 88.0 | 87.9 | 88.0 | 88.3 | 88.9 | 89.8 |
| 1 | 0.75 | 0.75 | 87.5 | 88.2 | **90.7** | 89.1 | 89.6 | 90.7 |
| 1 | 1 | 1 | **88.8** | **89.1** | 89.7 | **89.1** | **89.9** | **90.8** |

---

[13]https://github.com/PyTorchLightning/lightning-bolts

**Number of Attention Heads**    As discussed in Transformer [17], multiple-head attention significantly improve the performance in various applications than single-head attention. To investigate the effect of multi-head attention on learning cross-modal token-wise correspondence (*i.e.*, CTA module), we conduct experiments on pre-training MGCA using 1, 2 and 4 attention heads. The results are shown in Table 5. It demonstrates that using more attention heads does not substantially improve the linear classification performance. Instead, we observe that single-head attention shows the best performance in 5 out of 6 settings. We think the potential reason might be single-head attention is enough to align cross-modal token representations. Thus, we use single-head attention in CTA module by default.

Table 5: Ablation results of number of attention heads (N).

| N | CheXpert (AUC) 1% | 10% | 100% | RSNA (AUC) 1% | 10% | 100% |
|---|---|---|---|---|---|---|
| 1 | **88.8** | **89.1** | **89.7** | 89.1 | **89.9** | **90.8** |
| 2 | **88.8** | 88.7 | 89.0 | 89.1 | 89.6 | 90.7 |
| 4 | **88.8** | 88.7 | 89.4 | **89.4** | 89.7 | 90.6 |

**Number of Prototypes**    Table 6 shows the effect of number of cross-modal prototypes (K) on linear classification performance. We examine three configurations of MGCA: $K = 250, 500, 750$. The empirical study shows that $K = 500$ achieve the best performance in 5 out of 6 settings. However, the performance of these three configuration is pretty close. We think hierarchical cross-modal clustering methods might be a potential direction to capture more abundant semantic structures. This is also a part of our future work.

Table 6: Ablation study on the number of prototypes (K).

| K | CheXpert (AUC) 1% | 10% | 100% | RSNA (AUC) 1% | 10% | 100% |
|---|---|---|---|---|---|---|
| 250 | 88.4 | 88.8 | 88.7 | **89.2** | 89.8 | 90.6 |
| 500 | **88.8** | **89.1** | **89.7** | 89.1 | **89.9** | **90.8** |
| 750 | 88.3 | 88.9 | 88.9 | 88.9 | **89.9** | 90.7 |

**Embedding dimension**    We study how different embedding dimension (d) influence the linear classification performance on CheXpert and RSNA dataset in Table 7. It shows that $d = 128$ achieves the best performance in 5 out of 6 configurations.

Table 7: Ablation study of embedding dimension (d).

| d | CheXpert (AUC) 1% | 10% | 100% | RSNA (AUC) 1% | 10% | 100% |
|---|---|---|---|---|---|---|
| 32 | 88.2 | 88.8 | 88.9 | 88.5 | 89.2 | 90.3 |
| 64 | 88.6 | **89.2** | 89.3 | 88.8 | 89.4 | 90.5 |
| 128 | **88.8** | 89.1 | **89.7** | **89.1** | **89.9** | **90.8** |
| 256 | 88.2 | 88.7 | 88.8 | 87.6 | 89.5 | 90.5 |
| 512 | 88.5 | 88.9 | 89.0 | 88.9 | 89.4 | 90.7 |

# D    Additional Visualization Results

**Visualization of Activated Regions by ViT**    In our CTA module, we re-weight each visual tokens with respect to their contribution to image-level representations. Figure 1 visualizes the weights of visual tokens obtained by ViT. Recall that the weights are obtained by averaging ViT's last-layer attention weights across multiple heads. Highlighted pixels represent regions with relatively high attention weights. It shows that ViT automatically learns to focus on critical regions by aligning cross-modal instance-level representations.

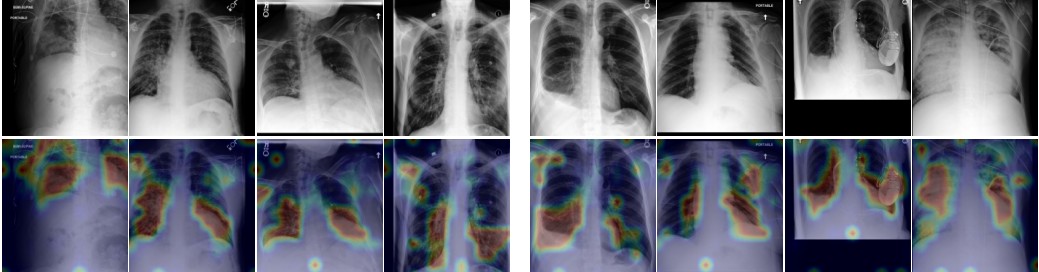

Figure 1: Visualization of activated visual tokens. Highlighted pixels represent important regions learned by Vision Transformer.

**Visualization of Relative Importance of Words**  Table 8 shows three examples of radiology reports. The top 10 important (*i.e.*, have the highest weights) words are shown in red font. Note that the weights of words are obtained by averaging BERT's last-layer attention weights across multiple heads. We observe that most of the highlighted words (*e.g.*, orogastric, opacities, atelectasis, effusion, pneumothorax) are closely related to patients' medical conditions, while most of the unrelated words (*e.g.*, the, is, are, with) have relatively smaller weights. Thus, BERT model learns to focus on disease-related words when aligning cross-modal instance-wise embeddings.

Table 8:  Visualization on attention weights of words learned by text encoder. Words with top 10 highest weights are highlighted by red font.

| |
| --- |
| orogastric tube seen with tip terminating in the stomach and last side port at the level of the ge junction recommend advancement of tube by 4cm a single frontal semi supine view of the abdomen demonstrates an orogastric tube with the tip terminating in the stomach and the last side port at the level of the ge junction there is a normal bowel gas pattern without evidence of ileus or obstruction there are metallic density sugical _ _ _ seen longitudinally traversing the midline there is hazy |
| overall improvement with decrease in right pleural effusion and right lung opacities hydropneumothorax has resolved pa and lateral views of the chest the small left pleural effusion has resolved the right pleural effusion has slightly decreased in size the right lower middle and upper lobe opacities have decreased there is mild linear right basilar atelectasis small loculated hydropneumothorax has resolved the left lung is clear the cardiac mediastinal and hilar |
| no evidence of acute cardiopulmonary disease the heart is normal in size the mediastinal and hilar contours appear within normal limits there is no pleural effusion or pneumothorax the lungs appear clear bony structures are unremarkable |