# OpenReview forum: "Multi-Granularity Cross-modal Alignment for Generalized Medical Visual Representation Learning"
_NeurIPS.cc/2022/Conference — NeurIPS 2022 Accept_

### Official Review · Reviewer_D1Wu · 2022-07-07

**Rating:** 7
**Confidence:** 5
**Soundness:** 4 excellent
**Presentation:** 4 excellent
**Contribution:** 3 good

**Summary:**

This paper proposes a Multi-Granularity Cross-Modal Alignment method, which learns data representations from medical scans paired with the corresponding text reports. The method exploits multiple unsupervised techniques to obtain the learned representations (e.g. contrastive losses, clustering with Sinkhorn-Knopp), and each of these techniques are utilized at the appropriate level of granularity.
The learned representations are then evaluated on a large set of image-based downstream tasks, to assess the quality of the image representations. The experimental results support the merits of the proposed method. However, there are some weaknesses and limitations, which will be listed below.

**Questions:**

It would be interesting to have a discussion in the paper on how the ITA level behaves with the existence of CPA. These two losses have a somehow contradictory behavior. The contrastive loss in ITA pushes the different instances (negatives) apart from positive instances. However, CPA clusters instances in the feature space in order to learn disease (prototype) clusters. CPA disregards the identity of the instance and rather tries to capture high-level unifying topic models.
How do they behave then? It seems to me in Table 4 that the addition of the CTA loss is essential for improved performance (rows 2 and 4). But adding CPA and ITA alone (in row 3), seems to perform a little worse than row 2.
Please discuss how do you imagine the losses are behaving.

**Limitations:**

The authors list the limitations adequately

**Strengths And Weaknesses:**

Strengths:
- The proposed method is novel, if viewed as a whole framework that employs the granularity of the features that are present in medical scans. However, if one views each level in the method independently, these levels are not novel per se. But since the proposed method aims to encapsulate these levels together, then I deem the method as novel. The only level that I would deem more novel is the CPA, which proposes to learn cross-modal disease prototypes.
- The evaluation experiments are extensive. The paper reports results on multiple dataset benchmarks, which include multiple types of tasks (classification, detection and segmentation). The work also evaluates two types of image encoders (resnet & ViT).
- The paper is clearly written, and the used language is well understandable.

Weaknesses:
Mostly, I believe the weaknesses in this work are in some missing pieces:
1- Missing relevant references from the literature. The following are some works that perform contrastive learning on medical images:
* Taleb, Aiham, et al. "ContIG: Self-supervised Multimodal Contrastive Learning for Medical Imaging with Genetics." Proceedings of the IEEE/CVF Conference on Computer Vision and Pattern Recognition. 2022.
* Chaitanya, Krishna, et al. "Contrastive learning of global and local features for medical image segmentation with limited annotations." Advances in Neural Information Processing Systems 33 (2020): 12546-12558.
* Han, Yan, et al. "Pneumonia detection on chest x-ray using radiomic features and contrastive learning." 2021 IEEE 18th International Symposium on Biomedical Imaging (ISBI). IEEE, 2021.
* Feng, Ruibin, et al. "Parts2Whole: Self-supervised contrastive learning via reconstruction." Domain Adaptation and Representation Transfer, and Distributed and Collaborative Learning. Springer, Cham, 2020. 85-95.
* Xu, Jiarui, et al. "GroupViT: Semantic Segmentation Emerges from Text Supervision." Proceedings of the IEEE/CVF Conference on Computer Vision and Pattern Recognition. 2022.
There are quite a lot of works that are similar along this line, would be great to cite the most relevant once from the medical imaging literature.
Also I believe some references related to the methodological part (mainly cross-attention) are missing, such as:
Chen, Yen-Chun, et al. "Uniter: Learning universal image-text representations." (2019).
Lu, Jiasen, et al. "Hierarchical question-image co-attention for visual question answering." Advances in neural information processing systems 29 (2016).

---

> ### Author Response · Authors · 2022-08-02
> **Response to reviewer D1Wu**
>
> We highly appreciate your insightful reviews and positive comments on the novelty of our framework, the extensive evaluation, and the paper presentation. Our responses are shown as follows.
>
> > Q1: Mostly, I believe the weaknesses in this work are in some missing pieces.
>
> *A*: Thank you very much for pointing out these relevant references! We have added and discussed these related references in the revised paper. Please feel free to take a look at it.
>
> > Q2: It would be interesting to discuss in the paper on how the ITA level behaves with the existence of CPA. These two losses have a somehow contradictory behavior. The contrastive loss in ITA pushes the different instances (negatives) apart from positive instances. However, CPA clusters instances in the feature space to learn disease (prototype) clusters. CPA disregards the instance's identity and tries to capture high-level unifying topic models. How do they behave then? It seems to me in Table 4 that the addition of the CTA loss is essential for improved performance (rows 2 and 4). But adding CPA and ITA alone (in row 3), seems to perform a little worse than row 2. Please discuss how do you imagine the losses are behaving.
>
> *A*: We think the joint optimization of ITA and CPA leads to a somehow balanced embedding space that **retains the local smoothness** of instances and **semantic structure** of the whole dataset. In the training stage, we optimize ITA to retain the property of cross-modal smoothness. This strategy has been proved effective in several previous works [1][2][3]. As discussed in [4], we think the local smoothness of instance features is significantly important for transferring to downstream tasks. However, this cross-modal smoothness ignore the semantic structure in the dataset. This is why we also incorporate CPA into our framework. The CPA can provide a constraint (or regularization) for the embedding space and encodes more high-level semantic information into it. Thus, the learned features can demonstrate more generalized and transferrable performance.
>
> For the results in Table 4, we notice that row 3 (ITA+CPA) is better than row 2 (ITA + CTA) in the CheXpert dataset while performing slightly worse in the RSNA dataset. We believe there might be two reasons leading to this phenomenon: (1) The downstream task of RSNA dataset (binary classification) is **easier** than CheXpert (multi-label classification). In this case, the high-level semantic structure learned by CPA does not significantly affect the downstream performance in RSNA dataset. (2) The disease-level semantic structure (prototypes) of MIMIC-CXR dataset are more **similar** to CheXpert than RSNA. Thus, the learned high-level semantic information might be more applicable to CheXpert than RSNA. This might also partly explain why CPA trained in MIMIC-CXR dataset is more powerful in CheXpert than RSNA.
>
> References:
>
> [1]. J. Li, P. Zhou, C. Xiong, and S. C. Hoi. Prototypical contrastive learning of unsupervised representations. arXiv preprint arXiv:2005.04966, 2020.
>
> [2]. Y. Guo, M. Xu, J. Li, B. Ni, X. Zhu, Z. Sun, and Y. Xu. Hcsc: Hierarchical contrastive selective coding. arXiv preprint arXiv:2202.00455, 2022.
>
> [3] Y. Li, P. Hu, Z. Liu, D. Peng, J. T. Zhou, and X. Peng. Contrastive clustering. In 2021 AAAI Conference on Artificial Intelligence (AAAI), 2021.
>
> [4]. Nanxuan Zhao, Zhirong Wu, Rynson WH Lau, and Stephen Lin. What makes instance discrimination good for transfer learning? arXiv preprint arXiv:2006.06606, 2020.

---

### Official Review · Reviewer_ZGvT · 2022-07-07

**Rating:** 8
**Confidence:** 3
**Soundness:** 4 excellent
**Presentation:** 4 excellent
**Contribution:** 3 good

**Summary:**

The authors present a visual representation learning approach based on contrastive learning of correspondences between visual radiographs and textual radiology reports at multiple levels of granularity using three different learning objectives. The first objective features standard cross-modal alignment between radiographs and their corresponding radiology report pairs, by which they enforce instance level agreement. The second objective features a cross-attention approach that enforces the alignment between the visual token embeddings (local representations obtained using the Vision Transformer) and aggregated text token representations (where the visual token is used as a query for attention-based aggregation of word token representations), and likewise, the alignment between text tokens and aggregated visual tokens. This enforces alignment between mutually informative local image regions in the radiograph and parts of the textual radiology report, and the authors argue that this region-level alignment improves performance of resulting visual representations on downstream task that involve dense predicitons. The final objective features alignment on a higher-than-instance level, by enforcing consistent cross-modal clustering of representations of corresponding visual and text data. This implicitly encourages the radiographs and reports that share high-level semanics to have similar representations, regardless of instance-level pairing and modality. Authors also demonstrate that pretraining on in-domain images (medical) is important for performance on downstream tasks, and that general-domain pretraining is not as effective.

**Questions:**

- How is CTA implemented in cases when ResNet50 is used as the image encoder? In what way are visual tokens obtained? To what degree would you say the difference in the way the tokens are obtained can account for the difference in performance between the ResNet50 and ViT variant of your approach?
- Does the use of CTA give an even bigger performance improvement in a segmentation or object detection setting (compared to classification)?

**Limitations:**

The authors list the lack of consideration of retrieval tasks as one of the limitations of their work. However, the outlined scope of their experiments is in my opinion reasonable. With regards to potentially negative societal impacts, the authors mention the possible use of sensitive data in their framework. This is a general concern when medical data is used and is therefore not particularly specific to their work.

**Strengths And Weaknesses:**

Strengths:
- The paper is clearly written and well structured.
- Implementation details and reproducibility info are clearly provided, with appendix containing additional ablation studies justifying parameter and design choices.
- Extensive experiments (both qualitative and quantitative) as well as ablation studies. The choice of downstream tasks appears to be justified given the stated focus on learning visual representations, and they seem diverse enough to put different individual aspects of their approach to the test. The comparison with the main competing approach, GLoRIA, seems fair, as the authors also evaluated a variant of their model that uses the same image encoder as GLoRIA as well as the same pre-processing step.
- The idea and implementation of the disease-level cross-modal alignment module in this setting, to the best of my knowledge, can be seen as novel.

Weaknesses:
- Lack of an ablation study involving individual learning tasks in a object detection or segmentation setting (as was done under the classification setting). Their approach does seem to more convincingly outperform GLoRIA in a object detection and segmentation setting (compared to the classification featuring the ResNet-50 encoder), but it would have been interesting to see an ablation study that explicitly shows how relevant, for instance, CTA is in a dense prediction setting.
- Lack of error bars may be a potential issue, since the differences in performance between their approach and baselines (in some settings) are sometimes small.
- This particular setup for contrastive learning based approach for visual representation learning (also in the context of paired radiographs and radiology reports), as well as the idea and motivation behind token-wise cross-modal alignment objective, at its core, seem to be inspired by ConVIRT and/or GLoRIA, so the work can, to a degree, be seen as an extension of existing work and not entirely novel.

---

> ### Author Response · Authors · 2022-08-02
> **[2/2] Response to reviewer ZGvT**
>
> > Q5: This particular setup for contrastive learning based approach for visual representation learning (also in the context of paired radiographs and radiology reports), as well as the idea and motivation behind token-wise cross-modal alignment objective, at its core, seem to be inspired by ConVIRT and/or GLoRIA, so the work can, to a degree, be seen as an extension of existing work and not entirely novel.
>
> *A*: Thank you for your comments. We are glad to restate our contribution of our proposed approach. (1) Our MGCA framework is designed **for the first time**, to leverage multi-granularity semantic correspondence for generalized medical visual representation learning to facilitate versatile downstream tasks. The whole framework design based on the prior knowledge of multi-granularity is not trivial and needs careful balance among each component to make full use of their complementary properties at three different levels to learn better visual representations. We thereby carefully design different unsupervised learning strategies (e.g., contrastive losses, cross-attention, clustering with Sinkhorn-Knopp). (2) As you mentioned, the design of disease-level cross-modal prototypical alignment (CPA)  module to leverage the high-level inter-subject relationship semantic correspondences by enforcing the cross-modal cluster assignment consistency is interesting and novel. To the best of our knowledge, this is the first time of its kind in cross-modality learning. (3) For the experimental design, as we know, **label-efficient learning** is significantly essential in the medical image field. Our proposed framework can effectively leverage multi-granularity cross-modal correspondence for medical image pre-training and significantly boost the performance of several downstream tasks even trained with only 1% annotated data. Based on the above three points, we believe that our work will bring new insights for medical imaging and machine learning researchers on the task of medical image and report processing, as well as assist the development of the relevant clinical (radiology) fields.
>
> References:
>
> [1]. S.-C. Huang, L. Shen, M. P. Lungren, and S. Yeung. Gloria: A multimodal global-local representation learning framework for label-efficient medical image recognition. In Proceedings of the IEEE/CVF International Conference on Computer Vision, pages 3942–3951, 2021.
>
> [2]. Tete Xiao, Mannat Singh, Eric Mintun, Trevor Darrell, Piotr Dollar, and Ross Girshick. Early convolutions help transformers see better.In NeurIPS, 2021.
>
> [3] Dosovitskiy, Alexey, et al. "An image is worth 16x16 words: Transformers for image recognition at scale." arXiv preprint arXiv:2010.11929 (2020).

---

> ### Author Response · Authors · 2022-08-02
> **[1/2] Response to reviewer ZGvT**
>
> We highly appreciate your insightful comments and positive feedback on the extensive evaluation, the novelty of our proposed framework, and the detailed implementation. We would like to address your questions below one by one.
>
> > Q1: Lack of an ablation study involving individual learning tasks in an object detection or segmentation setting (as was done under the classification setting). Their approach does seem to more convincingly outperform GLoRIA in a object detection and segmentation setting (compared to the classification featuring the ResNet-50 encoder), but it would have been interesting to see an ablation study that explicitly shows how relevant, for instance, CTA is in a dense prediction setting.
>
> *A*: Thanks for your constructive suggestion. We have conducted an extra ablation study to validate the importance of CTA under dense prediction settings (medical image segmentation) on the SIIM dataset. The results are shown as follows.
>
> | | Training tasks |  |      | SIIM (Dice) | |
> | :--: | :--: | :--: | :--: | :--: | :--: |
> | ITA | CTA | CPA |  1%  | 10%  | 100% |
> | &#10004; | | | 25.0 | 43.2 | 59.9 |
> | &#10004; | &#10004; | | 47.6 | 55.4 | 61.3|
> | &#10004; | | &#10004; | 37.4 | 46.7 | 55.0 |
> | &#10004; | &#10004; | &#10004; | **49.7** | **59.3** | **64.2** |
>
> It is observed that CTA and CPA can both improve semantic segmentation performance when combined with ITA. When we train ITA, CTA, and CPA jointly, it achieves the best performance.
>
> > Q2: Lack of error bars may be a potential issue, since the differences in performance between their approach and baselines (in some settings) are sometimes small.
>
> *A*: Thanks for your suggestions. We have re-run the downstream task experiments (linear classification) of our methods three times and calculated the mean and standard deviations.
> We showed the error bars of our experiment results in the following table.
>
> |  | 1% | 10% | 100% |
> | :--: | :--: | :--: | :--:|
> | CheXpert (AUC) | 88.7 $\pm$ 0.18 | 89.13 $\pm$ 0.16 | 89.5 $\pm$ 0.21 |
> | RSNA (AUC) | 89.03 $\pm$ 0.11 | 89.92 $\pm$ 0.14 | 90.77 $\pm$ 0.04 |
> | COVIDx (ACC) | 73.9 $\pm$ 0.64 | 84.75 $\pm$ 0.22 | 92.85 $\pm$ 0.50 |
>
> According to the results, we notice that the error bar is pretty small relative, which shows that our proposed method performs stably in these downstream tasks.
>
>
> > Q3: How is CTA implemented in cases when ResNet50 is used as the image encoder? In what way are visual tokens obtained? To what degree would you say the difference in the way the tokens are obtained can account for the difference in performance between the ResNet50 and ViT variant of your approach?
>
> *A*: As indicated in GLoRIA[1], when ResNet50 is used as the image encoder, we take the feature maps ($f \in \mathbb{R}^{1024\times19\times19}$) of 3-rd bottleneck building blocks as token-level features. Then, we reshape the token-level features into $f \in \mathbb{R}^{1024\times361}$, where $361$ is the total number of image regions.
>
> We think the different network architecture, instead of the token acquisition strategy, mainly contribute to the performance difference between ResNet50 and ViT variants of our approach. In [2], the authors find that using a convolutional stem in ViT achieves much better performance than the ResNet50 counterpart, which also supports our thought that the difference of architecture design plays an important role in the performance gap. Specifically, ViT-based methods can retain more spatial or global information than ResNet [3], thus it achieves better performance.
>
> > Q4: Does the use of CTA give an even bigger performance improvement in a segmentation or object detection setting (compared to classification)?
>
> *A*: The use of CTA can give a larger performance improvement in a segmentation setting. The specifical ablation study can be seen in Q1.  In general, both CTA and CPA could enhance the dense prediction performance of downstream tasks.

---

> > ### Comment · Reviewer_ZGvT · 2022-08-08
> > **Response to the Authors**
> >
> > Thank you for the response. I appreciate the extra ablation study regarding the dense prediction task. It is indeed an interesting finding that further corroborates the claims made in the paper. I would suggest finding a way to include it in the eventual camera-ready version.
> > The error bars computed using the approach you mentioned seem to be significantly lower than the margin by which your approach outperforms GLoRIA, so I would not hold it against the performance itself, however, as a general principle, you should include error bars in results tables, particularly when comparing against other works.
> > I am inclined to agree on the novelty of the disease-level cross-modal prototypical alignment module, as well as the overall framework, and that the combination of the three components is not trivial from a technical standpoint.
> > Otherwise, thank you for answering the other questions. I have also noted the additional ablation performed at the request of reviewer RZTv, and the reply regarding the comparison with existing works.
> > I am inclined to keep the original rating.

---

> > > ### Author Response · Authors · 2022-08-08
> > > **Appreciate your feedback**
> > >
> > > Dear reviewer ZGvT,
> > >
> > > Thanks a lot for your time and valuable feedback! We will include your suggested experimental results in our paper.

---

### Official Review · Reviewer_RZTv · 2022-07-11

**Rating:** 5
**Confidence:** 4
**Soundness:** 2 fair
**Presentation:** 3 good
**Contribution:** 2 fair

**Summary:**

This paper presents a novel Multi-Granularity Cross-modal Alignment (MGCA) framework to seamlessly leverage the multi-granular semantic correspondences between medical images and radiology reports for generalized medical visual representation learning. Extensive experimental results on different downstream datasets show that the proposed framework achieves substantial performance with limited annotated data.

**Questions:**

(1) The authors incorporate several well-known technologies, including instance-wise alignment, fine-grained token-wise alignment, and disease-level alignment, into contrastive learning to enhance the generalizability of learned visual representations. In this case, which parts are new? Thus, the overall novelty needs to be strengthened.
(2) In Table 1, it can be seen that the proposed model performs better than other methods. However, because the proposed model has been pre-trained on a large-scale medical image and report dataset, whether other methods have been trained on the large dataset?
(3) The generated visual and text token embeddings are projected into normalized lower-dimensional embeddings, and the dimension d is set to 128. How to set the dimension? The parameter study should be included.
(4) From the results in Table 1, it can be seen that the proposed model obtains large improvements over DSVE and VSE++. Please the reasons for these situations.
(5) It is better to provide more discussions on using multi-granularity cross-modal alignment.

**Ethics Review Area:**

["I don’t know"]

**Limitations:**

See weakness.

**Strengths And Weaknesses:**

Strengths:
(1) A multi-granularity cross-modal alignment framework is proposed for learning generalized medical visual representations from free-text radiology reports.
(2) The overall structure is clear.
(3) The experimental results show the effectiveness of the proposed model.

Weaknesses:
(1) The authors incorporate several well-known technologies, including instance-wise alignment, fine-grained token-wise alignment, and disease-level alignment, into contrastive learning to enhance the generalizability of learned visual representations. In this case, which parts are new? Thus, the overall novelty needs to be strengthened.
(2) In Table 1, it can be seen that the proposed model performs better than other methods. However, because the proposed model has been pre-trained on a large-scale medical image and report dataset, whether other methods have been trained on the large dataset?
(3) The generated visual and text token embeddings are projected into normalized lower-dimensional embeddings, and the dimension d is set to 128. How to set the dimension? The parameter study should be included.

---

> ### Author Response · Authors · 2022-08-02
> **[2/2] Response to reviewer RZTv**
>
> > Q4: From the results in Table 1, it can be seen that the proposed model obtains large improvements over DSVE and VSE++. Please explain the reasons for these situations.
>
> *A*: We think the main reason for this situation is the different learning objectives. DSVE and VSE++ optimize a triplet ranking loss to learn image-text alignment. These methods only focus on minimizing the distance between the representations of true image and text pairs, without exploring the relationship with other samples. Therefore, as mentioned in [2], when applying the triplet ranking loss to medical images with high inter-class visual similarities, these methods can easily overfit by learning irrelevant patient/case-specific visual cues. These misleading cues will degrade its performance when transferring to downstream tasks.
>
> Differently, instance-wise contrastive learning (including ConVIRT, global loss of GLoRIA, and ITA of our method), optimizes an InfoNCE loss to learn instance-wise alignment, which contrasts positive image-text pairs and negative image-text pairs. As mentioned in relevant papers [4], InfoNCE maximizes the mutual information between true image-text pairs, which enables the image encoder to learn more transferable features.
>
>
> > Q5: It is better to provide more discussions on using multi-granularity cross-modal alignment.
>
> *A*: Our motivation for this paper is to leverage **naturally existing multi-granularity semantic correspondence** between medical images and radiology reports to learn better medical visual representations. As illustrated in Q1, we notice that this problem is important while challenging in the medical image domain.
>     To fill this gap, we are the **first** one to propose the MGCA framework and achieve **state-of-the-art performance** on several downstream tasks (i.e., classification, object detection, and semantic segmentation) with limited annotated data. Ablation study results in Table 4 also support our idea that the multi-granularity alignment significantly boosts the performance of learned medical visual representation.
>
> References:
>
> [1]. Y. Zhang, H. Jiang, Y. Miura, C. D. Manning, and C. P. Langlotz. Contrastive learning of medical visual representations from paired images and text. arXiv preprint arXiv:2010.00747, 2020.
>
> [2]. S.-C. Huang, L. Shen, M. P. Lungren, and S. Yeung. Gloria: A multimodal global-local representation learning framework for label-efficient medical image recognition. In Proceedings of the IEEE/CVF International Conference on Computer Vision, pages 3942–3951, 2021.
>
> [3]. Zhirong Wu, Yuanjun Xiong, Stella Yu, and Dahua Lin. Unsupervised feature learning via non-parametric instance discrimination. In CVPR, 2018. Updated version accessed at:
> https://arxiv.org/abs/1805.01978v1.
>
> [4]. He, H. Fan, Y. Wu, S. Xie, and R. Girshick. Momentum contrast for unsupervised visual representation learning. In Proceedings of the IEEE/CVF conference on computer vision and pattern recognition, pages 9729–9738, 2020.
>
> [5]. A. v. d. Oord, Y. Li, and O. Vinyals. Representation learning with contrastive predictive coding. arXiv preprint arXiv:1807.03748, 2018.

---

> ### Author Response · Authors · 2022-08-02
> **[1/2] Response to reviewer RZTv**
>
> We sincerely thank you for the comments. We would like to address your questions below one by one.
>
> > Q1: The authors incorporate several well-known technologies, including instance-wise alignment, fine-grained token-wise alignment, and disease-level alignment, into contrastive learning to enhance the generalizability of learned visual representations. In this case, which parts are new? Thus, the overall novelty needs to be strengthened.
>
> Thank you for your comments. We would like to highlight the overall novelty of our proposed approach from the following three aspects.
>
> - i.  Our MGCA framework is designed **for the first time**, to leverage multi-granularity semantic correspondence for generalized medical visual representation learning to facilitate versatile downstream tasks. For the novelty, our whole framework design based on the prior knowledge of multi-granularity is not trivial and needs careful balance among each component to make full use of their complementary properties at three different levels to learn better visual representations. We thereby carefully design different unsupervised learning strategies (e.g., contrastive losses, cross-attention, clustering with Sinkhorn-Knopp), instead of simply incorporating existing techniques on the contrastive learning paradigm. Contrastive learning is only one of the manners for achieving cross-modal alignment in our framework. Such framework novelty **is acknowledged** by reviewers ZGvT and D1Wu.
>
> - ii. We design a novel disease-level cross-modal prototypical alignment (CPA)  module to leverage the high-level inter-subject relationship semantic correspondences by enforcing the cross-modal cluster assignment consistency. To the best of our knowledge, this is the first time of its kind in cross-modality learning. Such CPA design novelty **is also acknowledged** by reviewers ZGvT and D1Wu.
>
> - iii. For the experimental design, as we know, **label-efficient learning** is significantly essential in the medical image field. Our proposed framework can effectively leverage multi-granularity cross-modal correspondence for medical image pre-training and significantly boost the performance of several downstream tasks even trained with only 1% annotated data.  Also, according to our ablation study in Table 4, these three different granularity correspondences all benefit downstream tasks and these benefits are complementary.
>
>     Based on the above three points, we believe that our work will bring new insights for medical imaging and machine learning researchers on the task of medical image and report processing, as well as assist the development of the relevant clinical (radiology) fields.
>
>
> > Q2: In Table 1, it can be seen that the proposed model performs better than other methods. However, because the proposed model has been pre-trained on a large-scale medical image and report dataset, whether other methods have been trained on the large dataset?
>
> *A*: Thanks for your comments. Our experiment comparison is fair and based on a similar pre-training strategy on large-scale medical image and report datasets. The methods in Table 1 are **all pre-trained** on two large-scale medical image and report datasets, as indicated by “pre-trained on CheXpert” and “pre-trained on MIMIC-CXR”. Note that the results of other methods on CheXpert and RSNA are from the original paper [1] [2]. The results of ConVIRT [1] on the RSNA dataset are from the reimplemented results in GLoRIA [2]. The results on COVIDx are implemented by ourselves.
>
> Also, for a fair comparison with the main competing approach GLoRIA, we also evaluated a variant of our model that uses the same image encoder and pre-processing step as GLoRIA did. This is also acknowledged by Reviewer ZGvT, thus our experimental comparisons are extensive and fair.
>
>
> > Q3: The generated visual and text token embeddings are projected into normalized lower-dimensional embeddings, and the dimension d is set to 128. How to set the dimension? The parameter study should be included.
>
> *A*: The embedding dimension d is set to 128, following the previous work [3][4]. Per your suggestion, we also conducted an ablation study on this hyperparameter and the results are shown in the following table. As we can see, d=128 performs better than other configurations in most of the settings.
>
> |      |      | ChexPert (AUC) | |  | RSNA (AUC)||
> | :--: | :--: | :--: | :--: | :--: | :--: | :--: |
> |   d  |  1%  | 10%  | 100% |  1%  | 10%  | 100% |
> |  32  | 88.2 | 88.8 | 88.9  | 88.5 | 89.2 | 90.3 |
> |  64  | 88.6 | **89.2** | 89.3 | 88.8 | 89.4 | 90.5 |
> | 128  | **88.8** | 89.1 | **89.7** | **89.1** | **89.9** | **90.8** |
> | 256  | 88.2 | 88.7 | 88.8 | 87.6 | 89.5 | 90.5 |
> | 512  | 88.5 | 88.9 | 89.0 | 88.9 | 89.4 | 90.7 |

---

> ### Author Response · Authors · 2022-08-04
> **Looking forward to any further discussions**
>
> Dear reviewer RZTv,
>
> Thank you for your valuable comments again. We have responded to your comments as below.
> We are looking forward to your feedback or any further discussion on your concerns.

---

### Meta-Review · Area_Chair_nvoh · 2022-08-26

**Recommendation:** Accept
**Confidence:** Certain

**Metareview:**

A multi-granularity cross-modal alignment framework is proposed, which learns data representations from medical scans paired with the corresponding text reports.
The reviewers find the appraoch novel and the paper well-written with an overall clear structure.
Extensive experimental results show the effectiveness of the proposed model and experimental details are provided.

After the discussion with the authors, all reviewers vote towards acceptance of the paper.

**Award:**

No

---

### Decision · Program_Chairs · 2022-09-14

Accept